# A Qualitative Study on Parental and Community Stakeholder Views of the Link between Full-Day Kindergarten and Health in Southern Nevada

**DOI:** 10.3390/children6020026

**Published:** 2019-02-08

**Authors:** Courtney Coughenour, Jennifer Pharr, Maxim Gakh, Sheila Clark, Prescott Cheong

**Affiliations:** School of Community Health Sciences, University of Nevada Las Vegas 4505 S. Maryland Pkwy, Box 3064 Las Vegas, NV 89154, USA; jennifer.pharr@unlv.edu (J.P.); maxim.gakh@unlv.edu (M.G.); sheila.clark@unlv.edu (S.C.); cheonp1@unlv.nevada.edu (P.C.)

**Keywords:** educational attainment, half-day kindergarten, social determinants of health, public health, qualitative research, health impact assessment, full-day kindergarten

## Abstract

Studies show that children who attend full-day kindergarten (FDK) experience both academic and developmental benefits compared to children who attend half-day programs. Sectors outside of health, such as education, can have important intended and unintended impacts on health. The purpose of this qualitative study was to understand perceptions of parental and other stakeholders in Southern Nevada (USA) about the education–health link, and to understand priorities regarding how FDK access could affect health. Two 90-minute focus groups were conducted with 14 adult stakeholder participants representing parents, current and former teachers, and community members. Transcripts were analyzed using conventional content analysis. Eight major themes and several subthemes emerged; findings related to each are discussed. ‘Access’ was mentioned most frequently (*n* = 43), followed by ‘Time’ (*n* = 25), and ‘Lifetime educational attainment’ (*n* = 17). Participants were overall in favor of expanding access to FDK and felt that FDK could improve social skills, increase the amount of physical activity, and provide additional time for educators to detect additional learning disabilities when compared to half-day programs. Although the purpose was to understand priorities related to the education–health link, participants spent little time discussing this, suggesting this association is not inherently considered. Health and education stakeholders should collaborate to increase awareness, as this link may serve as an upstream approach to downstream effects on population health outcomes.

## 1. Introduction

The benefits of high-quality, early childhood education programs on low-income youth are well documented [1,2,3,4]. Findings indicate that children who attend full-day kindergarten (FDK) experience both academic and developmental benefits compared to children who attend half-day programs. In the short term (at the end of kindergarten), it appears that most students benefit from FDK regardless of race/ethnicity, income, or knowledge of English [2,3,4,5]. In the long term, low socioeconomic status, minority, English Language Learner (ELL), and inner-city students maintain significant differences in math and reading test scores in the third and fifth grades, compared to similar populations who did not attend FDK [2,3]. However, there is some controversy surrounding FDK. In a two-year study of a midwestern kindergarten program, Elicker and Mathur (1997) reported that ‘Proponents of full-day claimed that a longer day allows for better assessment of children’s educational needs, more time for individualized instruction, a broader, more developmentally appropriate curriculum, less stress for teachers and children, and needed child care relief for full-time working parents’ [6]. In contrast, opponents of FDK cite costs of the program [6,7] and increased stress and burn-out of young children due to the long instruction [6]. However, research does not conclusively substantiate the claims that kindergarten-aged children experience burnout in FDK [5,8].

In the USA, about 77.1% of students enrolled in kindergarten attend FDK and about 22.9% attend half-day kindergarten (HDK) [9]. While there is variation across states’ kindergarten requirements and programs, a typical half-day consists of about 3 h of instruction and a full-day consists of 5 to 6 h of instruction [10].

Partly because many education policies and related financial supports occur at the state level, there is variability across states in how kindergarten programs are structured. Some states require that HDK be offered, some require that FDK be offered, while some have no requirements. For example, the state of Alaska does not require school districts to offer kindergarten at all, Maryland requires FDK to be offered but does not require districts to offer HDK, and Rhode Island requires HDK to be offered but does not require districts to offer FDK [11].

Nevada consistently ranks low on assessments of educational outcomes based on test scores, graduation rates, and other educational metrics (see Table 1). State and local decision-makers, stakeholders, and community members frequently consider ways to improve the K–12 education system through education-oriented initiatives. One of these initiatives has focused on FDK, which at the time of this study was not universally available in Nevada’s public schools. Public schools in the state could, but were not required to, offer FDK. In some schools, FDK was available free of charge to parents, while in others, a combination of HDK and family-paid tuition FDK were available. Yet in others, only HDK was available with no option for FDK. The qualitative study discussed within this manuscript was conducted as part of the scoping phase of a health impact assessment (HIA) of FDK in Nevada [12]. One of the goals of scoping in an HIA is to identify what health effects to address and considering input from stakeholders is essential. The results from this focus group informed the larger HIA.

While most people agree that medical care and healthy behaviors impact health, the environmental and social determinants of health are equally, if not more, important [13,14,15,16]. Education is one of several social determinants of health. Ultimately, sectors outside of health, such as education, can have important intended and unintended impacts on health. The general link between education and health is well established. In the aggregate, those who are more educated live healthier and longer lives. They engage in more health-promoting behaviors such as meeting the recommended minutes of physical activity and consuming a healthier diet. They also engage in fewer health-compromising behaviors such as using tobacco [17,18]. Those who are more educated suffer from lower rates of both chronic and acute diseases [17,18]. However, the relationships between specific education programs and policies on one hand and health determinants and outcomes on the other are less understood. The connections between education and health are more complicated than access to goods and services and individual health behaviors and understanding this link must also take into account the social determinants of health and health equity. The purpose of this qualitative case study was to gain insight into what parents and other community stakeholders in Southern Nevada (USA) thought about the link between education and health, and to understand their priorities regarding how access to FDK could affect health.

## 2. Materials and Methods

### 2.1. Participants

Participants were recruited via email through several existing listservs which included all registered emails with the Nevada Parent and Teacher Association, Honoring Our Public Education (a Southern Nevada parent advocacy group), the University of Nevada, Las Vegas/Consolidated Students University of Nevada preschool, Las Vegas Downtown Achieves (a collective organization of public, private, and nonprofit sectors focused on the academic success of students in downtown Las Vegas), and The Guinn Center (stakeholders interested in state-level policy). While the actual number of unique emails sent out is unknown, it was likely that the number was large considering that the Clark County School District is the fifth largest in the nation, with over 320,000 enrolled youth [19]. There were 18 confirmative responses to the event, and a total of 14 participants who ultimately attended the event representing stakeholders that included parents, current and former teachers, and community members. While stakeholders self-identified themselves into these categories at the focus group, we did not document the information in written format and cannot accurately estimate the number of participants in each category. Participants over the age of 18 years were offered food and a $25 gift card for participating. Informed consent was obtained from all individual participants included in the study. This project was given Exempt status from the University of Nevada, Las Vegas Office of Research Integrity.

### 2.2. Location

The focus group was held after hours (5:45 p.m.) at a local preschool. Attempts were made to enhance convenience and minimize barriers to attendance. The preschool was centrally located in town, was accessible by public transit, and free parking was available. Parents were invited to bring their children as a replacement for the need to arrange for childcare.

### 2.3. Procedure

Two 90-minute focus group meetings were conducted with different sets of stakeholders. Focus groups were started simultaneously with the same brief 10-minute presentation that explained the current availability and funding structure of kindergarten in Nevada, as well as a brief explanation of how proposed policy may impact this structure. A brief explanation of the link between education and health was given, which included some examples of short- and long-term health effects (i.e., physical education augmenting physical activity behaviors, and educational attainment resulting in fewer negative health behaviors such as smoking). The 14 stakeholders were then split into 2 groups that each contained 7 participants. Discussion for each group was facilitated by two team members. Focus group discussions were based on open-ended questions related to the relationship of FDK with education and health outcomes, the general relationship between health and education, and funding options for FDK (see Appendix A for questions). All focus groups were conducted in English. Both focus group sessions were voice-recorded. They were subsequently transcribed (as close to verbatim as possible) by a research assistant.

### 2.4. Data Analysis

Using conventional content analysis, transcripts from each focus group were analyzed using open coding and axial coding [20]. Two research assistants participated in the coding process. First, they independently read through the focus group transcripts several times for familiarization. Next, the research assistants completed individual open-coding analyses in which the material was broken down into broad concepts and categories. Then, they performed axial coding and clustering by arranging the codes into themes to better reveal the associations and meanings behind the open codes [20]. Upon completion of independent analysis of the data, the two researchers negotiated themes, subthemes, and coding of the transcripts, discussing and mitigating any discrepancies. Lastly, selective coding was used to pick out quotes that exemplified each code and theme.

## 3. Results

Eight major themes emerged from our qualitative analysis of the transcripts. Within these major themes, several subthemes were also identified. All of the themes and subthemes are listed in Table 2.

The major theme of ‘Access’ was mentioned most frequently (*n* = 43) by the participants in each focus group. Participants expressed a desire for FDK to be mandatory for all students and funding for kindergarten to be consistent across schools. In addition, a concern expressed by study participants was the lack of appropriate FDK for students with individualized education programs (IEPs) or who were diagnosed as in need of special education services. Representative quotes illustrating these themes are listed below.

Q1: *It [FDK] should definitely be straight across the board, which maybe not, it should probably be mandatory, not really optional*.(Group A)

Q2: *I think that they should mandate that all kids go to FDK because then you have the difference of education, when children enter the kindergarten classroom… they can find the funding for the kids to go to FDK, which I think is important*.(Group B)

Q3: *So, in special kindergarten, basically, are these children who have IEPs and they need special attention…and my daughter is, you know, well, they were like she’s borderline and we should send her to that, but that’s when they offered half-day…*(different focus group participant speaking)*…I was told the same thing; I said I have to put him in kindergarten that offers all-day, and they said if he goes to special kindergarten, it’s only half-day*.(Group A)

Another theme repeatedly discussed by participants involved schools’ Title I status. Some participants expressed the belief that the process to determine designation of Title I status was unjust and unfair because Title I status was determined solely by one’s home address. Participants stated that Title I schools offered additional and needed services and benefits for students, including more comprehensive education, such as IEPs, greater parental involvement, access to free FDK, and, in some cases, funding for pre-K and supplementary learning programs. Specifically, participants discussed examples of families who wanted to enroll their child in FDK, but because they were not residing in a Title I school catchment area and lacked access to free full-day programs, they were unable to send their kids to FDK due to costs.
Q1: *I mean, cause, there’s only so many Title I schools in Nevada, right…So, for the parents that live in that area, they have the luxury, you know, of having their kid in school all day as opposed to someone who lives outside of the area like myself, I don’t have that luxury. So, I’m either putting them in half-day [kindergarten] or paying for tuition which, you know, I can’t afford the cost…*(Group A)
Q2: *I think, like you said with the Title I schools, sometimes there’s benefit in that they focus more on teaching the whole atmosphere …*(Group B)
Q3: *In the Title I program that my son went to last year…there were a lot of programs because they get so much funding…Fire safety was one of them. He came home and taught me, you know, he knew to dial 9-1-1 in case of an emergency…*(Group A)

It is interesting to note that participants articulated the importance of additional funding to be directed toward improving teacher salaries and providing teachers with additional classroom assistance. Overall, they also stated that changes in funding would have a large effect on health and development.
Q1: *Teachers need an incentive to, like, be a kindergarten teacher because some teachers do leave the district. They leave the district to go to another state or another city or something where they offer better competitive rates…*(Group A)
Q2: *[FDK] has some specific curriculum items for healthy habits, if that’s brushing your teeth or getting some exercise every day. There’s some particular curriculum items...*(Group B)
Q3: *A lot of times when kids are at home, the parents are busy working and they don’t really have that chance to play. So having the kids at school, and they have PE [physical education class], that really makes a big difference...*(Group B)

The second most common major theme identified was ‘Time’ (*n* = 25). Study participants were universally critical of the truncated school schedule created by HDK. In particular, participants believed that HDK would be a detriment for children, as it does not allow enough time for students to practice the necessary foundational educational skills. They also believed that the condensed, half-day school schedule was more stressful for children, compared to a full-day schedule. Participants’ comments suggested a belief that FDK would allot the necessary amount of time required for students to practice basic academic skills.
Q1: *I don’t think that half-day, two hours and fifteen minutes, are cutting it anymore… they want them reading and learning sight words and, already, addition, you know? (Teachers) at the school only get two hours with them, it’s not enough…*(Group A)
Q2: *They don’t do a lot in the two and a half hours. They don’t get a nutritional break, you know, a time to just go and play. This is strictly business…that’s stressful to me…*(Group B)

Moreover, focus group B reported that HDK was problematic for parents, the chief concern being the difficulties associated with coordinating childcare for the remainder of the day. ‘Cost to Families’ (*n* = 11), another major theme in our analysis, compounded the issues associated with ‘Time’ by adding another element for parents to consider when sending their children to HDK. These issues were magnified among parents with full-time work schedules. Lastly, participants reported that a consequence resulting from the logistical struggles in managing the half-day schedule is that parents often opt to keep their children enrolled in daycare for an extra year instead of enrolling them in kindergarten because it is more accommodating for their schedules. This is an option in Nevada since kindergarten is not mandatory and a child can be enrolled directly into first grade if they pass a test for proper grade level placement.
Q1: *The way it’s set up now where it’s [half-day kindergarten] only two hours or two and a half hours, it’s actually creating either another barrier because now you have to find someone to pick the child up at a certain time, and it’s an awkward timeframe within the day. And then it proposes another cost because then you have to find someone else to watch them if you’re working full-time or whatever…*(Group B)
Q2: *Most of them [parents] have to [use daycare rather than HDK] because they work full-time and it just poses too much of a problem to pick them up at 10:30 as opposed to when they know they can drop them off at daycare and they can leave them all day. And, although they didn’t anticipate having to go an extra year [to daycare], but you just don’t have a choice…*(Group B)

‘Community and School Involvement’ (*n* = 23) ranked as the third most common theme. The majority of study participants noted they had some type of direct involvement with the Clark County School District (CCSD), either being a parent of students who graduated from or were currently enrolled in CCSD or currently or previously working for CCSD in some capacity.

The subtheme that garnered the greatest amount of dialogue was ‘Lack of communication between school and parents’. Focus group participants emphasized that they felt as if there was limited interaction with parents about student achievement and progression, with lack of communication increasing in middle school. In contrast, it is interesting to note that one of our study participants, who was a teacher, mentioned that parental involvement decreased sharply starting in third grade. Participants stated that community and school involvement contribute to student success.
Q1: *Well, in my experience, and I have a son in seventh grade, it started in middle school, I mean in elementary, they’re always communicating with you with what’s going on and all of a sudden they get to sixth grade and “splat!”…they don’t want to talk to you in middle school…*(Group A)
Q2: *With community involvement, it is so strong that in some schools you see massive improvements when that takes place…to me education is a triangle; you’ve got the student, the school with the teachers, and the parents. And, when something is pulled from the triangle, it’s not connecting, and so it’s going to throw it off somewhere…*(Group B)

The experiences and beliefs expressed by the focus group members revealed a sense of a fragmented relationship between the parents and schools. One suggestion that was brought up during Focus Group B discussions was a revision to the current ‘report card’ system. Study participants expressed their desire for report cards to provide more in-depth information on the academic progress of students. However, they subsequently shared anecdotes of a failed pilot program attempted by certain schools to amend the traditional ‘ABC report card system’. Given that this subject received substantial discussion, it may be useful for school officials to reexamine, as it does hold the potential for encouraging interaction between parents, teachers, and students.

Q1: *This year they switched back. Our principal could have kept going with the new “approaches standards, meets standards, and exceeds standards”, but she felt that since the whole district was not changing over to that, and…that the change is not consistent to middle school, she decided, even though she believed in the other one [approaches, meets, exceeds standards], she decided to go back to the “ABC”. Coming home, I’ve got the first report card and I’m like, ‘it’s so minimal.’ It’s like it’s the letter grade on this assignment, so what does that assignment mean? You know, it’s, you’re not getting the standards…*(Group B)

‘Health’ (*n* = 16) was also a major theme that emerged from the focus group discussions. This was expected, as the purpose of this qualitative study was to focus on the link between education and health, with many of the moderator questions pertaining directly to health. Focus group participants indicated that benefits from FDK would be experienced largely in the social and behavioral aspects of health. They specifically pointed to improvements in social skills and overall manners in the classroom. Also, participants reported that physical health was boosted through FDK because of the increased play and exercise. Participants acknowledged that the added class time in FDK would enable more opportunities for teachers and school officials to identify students with potential learning disabilities.
Q1: *So my four–year-old who went to elementary last year, where … the school program was from 7 to 2:16, it was great. He learned a lot of things that he wouldn’t have been able to learn at home. On top of that, he gained a lot of social skills hanging out with other students his age…*(Group A)
Q2: *The physical development (would be enhanced) having the kids at school and they have P.E. That really makes a big difference...*(Group B)
Q3: *What about for children with IEPs …? My son, he’s four, he has an IEP for a developmental delay. I mean with all-day-kindergarten, they more than likely will pick up more students that need IEPs or have to be referred…*(Group A)

Two other subthemes merit attention due to their relatively high frequency of appearances. One is ‘FDK Improves Lifetime Educational Attainment’ (*n* = 17). Focus group participants emphasized the importance of FDK in building foundational skills as compared to HDK and noted that they believed it would lead to higher college achievement and lower dropout rates in Nevada.
Q1: *When you build a strong foundation, it’s kind of like a pyramid, everything else has got to lie on a rock on the base. When you build a strong base, then it’s easy to scaffold and put things on top of each other…and that’s why it’s so important to have an FDK…*(Group A)
Q2: *I think it [FDK] would improve the dropout rate. We’re like one of the highest; it would probably definitely decrease the dropout rate…*(Group B)

Under the theme of ‘Limited Classroom Resources’ (*n* = 14), the subtheme of ‘Overcrowded classrooms’ warrants mentioning. The majority of issues discussed within this subtheme were related to the challenges faced by teachers in managing large classroom sizes.
Q1: *So, if they do this, they need to decide how many [students] can be [in a class] and what’s the ratio? I mean, if you’re going to open an FDK but you’re going to have 37 kids in the classroom, everything we’re trying to work for, it’s not going to work…*(Group A)

Another education-related theme discussed (*n* = 13) had to do with expectations placed on parents, students, and teachers. Of the three subthemes, participants were most concerned with high academic expectations placed on kindergarten students. 

## 4. Discussion

The purpose of this qualitative case study was to gain insight into what parents and other community stakeholders thought about the link between education and health, and to understand their priorities regarding which health issues could be improved with expanded access to FDK. Interestingly, participants did not spend a lot of time discussing the education–health link, although many of the open-ended moderator questions were directly related to this topic. However, the fact that ‘Health’ ranked fifth out of eight major themes is telling. Access was the most discussed theme, although there was only one moderator question directly linked to this issue. It is possible that participants were most interested in this issue because of the attention that access to FDK was receiving at the time, including some local media coverage. It is also possible that participants self-selected to participate in the focus groups given their interest in education in general, and potentially FDK in particular. It is likely that these issues had an influence on the focus group discussions, but also highly likely that focus group participants did not inherently consider the association between education and health when thinking about FDK. If the latter is true, it would be beneficial for both health and education stakeholders to collaborate on increasing the awareness of the education and health link, as it has the ability to serve as an upstream approach to downstream effects on population health outcomes [13,14,15,16]. Although the health benefits of educational attainment are well documented in public health research, the association between access to FDK and health is not. More research, including longitudinal studies, is needed to understand this relationship. It is possible that additional research demonstrating this relationship would alter perceptions about the connections between FDK and health.

Another interesting finding was that participants felt that the condensed half-day school schedule was more stressful for children, compared to a full-day schedule. This finding differs from that of Elicker and Mathur (1997), who reported opposite parental concerns, that full-day schedules might add stress to children and result in a burnout [6]. This finding is promising given that research has disproven a burnout effect [5,8], and may indicate one less controversy related to FDK. Additionally, participants voiced that HDK was more stressful for parents in terms of time and familial cost of either getting their child to daycare after or before HDK or having to pay for FDK. 

With regard to the health impacts of FDK, participants stated beliefs that FDK could improve social skills, increase the amount of physical activity, and provide additional time for educators to detect additional learning disabilities when compared to HDK. This is similar to findings from Brannon’s interviews with parents who felt that their children who attended FDK benefited from socialization and peer relations, adjusted better to the first-grade schedule, and had more time to be active through play compared to their children who attended HDK [21].

Though it is recommended that sample sizes for focus groups remain small, this study included a total of 14 participants and was a voluntary convenience sample. Many participants were proponents of FDK access, and some even suggested making it mandatory, with no opinions shared that were in opposition to FDK. Yet previous research has found both proponents and opponents of FDK. It is unclear if the sentiment surrounding full-day programs is shifting, or if our findings were due, in part at least, to a sampling bias whereby those most passionate about education or FDK agreed to participate in the study. It is possible that current findings may not be generalizable to a larger population. The issues surrounding FDK access and funding span statewide, but our findings are just those from stakeholders in Southern Nevada. It is unknown if findings would have differed if focus groups were conducted state-wide. Additionally, perceptions and priorities related to the education–health link may differ outside of the USA. Thus, it is possible that the transferability of current findings may not be relevant to other contexts [22]. Validity and reliability are important in qualitative research; to ensure dependability [22], this study relied on focus group methodology to collect data and an accepted method of content analysis for application, all of which are documented in detail in this manuscript. To enhance reliability and credibility [22], two researchers employed the same method of analysis across focus group content, and both coders discussed and mitigated discrepancies. While focus groups are one methodology of collecting experiential data, it may be that individual interviews with stakeholders would have revealed experiences that were not part of the focus group discussion.

## 5. Conclusions

This study examined what parents and other community stakeholders thought about a somewhat controversial issue, full-day kindergarten (FDK). Overall, the participants in this study favored FDK over half-day kindergarten (HDK) and perceived access to free FDK as having educational, health, and community benefits. As researchers across disciplines, including health and education, increasingly recognize the connections between different sectors, it is critical to understand the attitudes, beliefs, and perceptions of stakeholders and community members about important issues with cross-sector implications. This case study helped illuminate some of these perceptions, although researchers felt that the current study may have lacked opinions from opponents of FDK. Study findings could be used by school administrators and other FDK advocates who wish to pursue efforts to expand access, as it may be less of a controversial issue as once thought. Additionally, increasing awareness of the link between education and health by all stakeholders is critical for the success of expanding access to FDK.

## Figures and Tables

**Table 1 children-06-00026-t001:** Nevada Student Test Scores and High School Graduation Rates Compared to USA.

National Exam: NAEP, 2011	Nevada	USA
Grade 4 Math, % proficient or above ^1^	32	38
Grade 4 Reading, % proficient or above ^1^	24	32
Grade 8 Math, % proficient or above ^1^	25	33
Grade 8 Reading, % proficient or above ^1^	22	29
High school graduation, % ^2^	70.7	81.4

^1^ National Center for Education Statistics (NCES), 2014; ^2^ US Department of Education, 2014.

**Table 2 children-06-00026-t002:** Major Themes and Subthemes Identified through Two 90-Minute Focus Groups (A and B) Conducted in Southern Nevada with 14 Parents and Other Community Stakeholders.

Major Theme *n* = Addition of Subthemes	Subthemes	Frequency Group A	Frequency Group B
Access (Total *n* = 43)	FDK should be available to all students	11	4
	Title I designations and associated services	3	4
	Funding increases needed for NV school system	7	3
	Reform costly school administration system	0	5
	Availability of IEPs	6	0
Time (Total *n* = 25)	HDK schedule not long enough to benefit students	7	4
	Parental difficulties in managing HDK schedule	0	9
	FDK schedule enables academic and ancillary learning	1	4
Community and School Involvement (Total *n* = 23)	Limited communication between school and parent	9	2
	Desire for changes in report card system/reporting on progress made by students	0	5
	Parent engagement/volunteering	3	0
	Benefits to parents and community members (in addition to students)	1	3
Lifetime Educational Attainment (Total *n* = 17)	FDK enables the foundational skills for higher educational attainment and employment	2	11
	FDK improves life skills through supplementary classes (i.e., nutrition)	2	0
	Lifetime educational attainment is hindered without preschool	0	2
Health (Total *n* = 16)	FDK impacts behavioral and social health	6	1
	FDK provides more opportunity for utilization of school services	4	1
	FDK impacts physical health	1	3
Limited Classroom Resources (Total *n* = 14)	Overcrowded classrooms	7	1
	Desire for classroom aides	6	0
Expectations (Total *n* = 13)	Academic expectations of kindergarten are rigorous	6	2
	Parental involvement in academics necessary to meet high standards	3	0
	Demands placed on teachers to enable students to meet standards are high; results in teacher burnout	2	0
Cost to Families (Total *n* = 11)	FDK is cost prohibitive (when required to pay for remaining half-day)	3	2
	Daycare is cost-prohibitive	1	3
	Ancillary costs associated with HDK	0	2

FDK = full-day kindergarten; HDK = half-day kindergarten; IEPs = individualized education programs; NV = Nevada.

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
