# Peer review of "A Qualitative Study on Parental and Community Stakeholder Views of the Link between Full-Day Kindergarten and Health in Southern Nevada"

_children, 2019, doi:10.3390/children6020026_

Reviewer 1 Report

The paper is interesting and it is presented in an appropriate and scholarly tone.The theoretical background could be broader. It could also include an overview of the health education currently provided in the kindergarten in Southern Nevada. Such information would also deepen discussion. The methodology is quite well described. The results are clearly presented and they are discussed in relation to previous studies. The reliability and validity of the study could be more in-depth to look at qualitative justification criteria. Below are my comments concerning the different chapters.

Abstract

Theoretical background of the study is stated very shortly but it illustrates the relationship between this research and previous studies in a sufficient way.

The purpose of the study is clearly stated.

What concerns on the material and methods they should be described more exactly. Also in the Abstract it should shortly be told:

- Where was the material gathered (e.g. country/state; city/rural area)?

- How was the material gathered?

- Who were the participants (e.g. the number of the participants and the age limit)?

- There are a wide variety of methods that are common in qualitative measurement. So, it should be stated what qualitative approach has been used. The used analysis method should also be mentioned.

The main results and the conclusions are described well.

Introduction

The theoretical background related to the research topic and questions has been largely explained. In addition, it would be necessary for international readers to indicate in the text the country in which the studies were conducted. (Especially, it matters when talking about percentages, for example; see p. 2, lines 49-50).  

From the point of view of international readers, it would be interesting to have an overview of the health education currently provided in the kindergarten in Southern Nevada. Such description would help to better understand the purpose and meaning of this study and support also implications in the Discussion chapter. It is therefore advisable to include such information in the Introduction.

In addition, it would also be interested to know what the parents and other stakeholders know about health education given in Kindergarten. However, this question was apparently not addressed in this study and will remain in the future.

Research material and methods

The acquisition of the participants and the methods of collecting and analyzing the material have been described quite well. However, it would also be good to say that the study is a case study and the approach was a content analysis. In addition, it would be good to state which kind content analysis approach it was used.  

Reliability and validity of the methods  and the related ethical perspectives should also be considered.

Results

The results are described well and they are interesting.

Discussion

Main results are clearly presented and justified by previous studies. The meaning of the study has been discussed. There has been presented some thoughts about the reliability and validity of the study, but it could be more in-depth to look at qualitative justification criteria (credibility, transferability, dependability, confirmability, see e.g. Lincoln, Y.S. & Guba, E.G. (1985). Naturalistic Inquiry. Newbury Park, CA: Sage Publications;Lincoln, Y.S.(1995). Emerging Criteria for Quality in Qualitative and Interpretive Research. Qualitative Inquiry, 1(3),275-289. http://ideanetworking.com.au/docs/interpretiveresearchmethods/Interpretive_Criteria-1995-Lincoln-275-89.pdf)

The tables are mainly ok, but an explanation for the abbreviation NV is missing from the Table 2. It should be added.

References are well chosen.

Author Response

The paper is interesting and it is presented in an appropriate and scholarly tone.The theoretical background could be broader. It could also include an overview of the health education currently provided in the kindergarten in Southern Nevada. Such information would also deepen discussion. The methodology is quite well described. The results are clearly presented and they are discussed in relation to previous studies. The reliability and validity of the study could be more in-depth to look at qualitative justification criteria. Below are my comments concerning the different chapters.

Abstract

Theoretical background of the study is stated very shortly but it illustrates the relationship between this research and previous studies in a sufficient way.

The purpose of the study is clearly stated.

What concerns on the material and methods they should be described more exactly. Also in the Abstract it should shortly be told:

- Where was the material gathered (e.g. country/state; city/rural area)?

Thank you, we have added Southern Nevada, USA to the abstract

- How was the material gathered? - Who were the participants (e.g. the number of the participants and the age limit)?

We have added this information to the abstract. It now reads “Two 90-minute focus groups were conducted with 14 adult stakeholder participants representing parents, current and former teachers, and community members.”

- There are a wide variety of methods that are common in qualitative measurement. So, it should be stated what qualitative approach has been used. The used analysis method should also be mentioned.

We have replaced qualitative analysis with “conventional content analysis.”

The main results and the conclusions are described well.

Introduction

The theoretical background related to the research topic and questions has been largely explained. In addition, it would be necessary for international readers to indicate in the text the country in which the studies were conducted. (Especially, it matters when talking about percentages, for example; see p. 2, lines 49-50).  

Thank you, we have added the USA in the introduction twice: Pg 2 when discussing the % of children enrolled in full and half day kindergarten, and in the statement of purpose.

From the point of view of international readers, it would be interesting to have an overview of the health education currently provided in the kindergarten in Southern Nevada. Such description would help to better understand the purpose and meaning of this study and support also implications in the Discussion chapter. It is therefore advisable to include such information in the Introduction.

The authors agree that this is interesting, but chose not to add a discussion on this to the paper because there are very few health standards required for kindergarten in Nevada. As a result, the health education in kindergarten varies from school to school (with over 200 elementary schools in Southern Nevada alone). Additionally, the study was most interested in understanding how stakeholders perceived the education-health link.

In addition, it would also be interested to know what the parents and other stakeholders know about health education given in Kindergarten. However, this question was apparently not addressed in this study and will remain in the future.

We agree that this would have been interesting to understand, and will hopefully be addressed in future research.

Research material and methods

 The acquisition of the participants and the methods of collecting and analyzing the material have been described quite well. However, it would also be good to say that the study is a case study and the approach was a content analysis. In addition, it would be good to state which kind content analysis approach it was used.  

Thank you for pointing this out. We have added that it is conventional content analysis to the abstract and methods section 2.4. We also added that it is a case study in our purpose statement (introduction section), and first sentence of the discussion.

Reliability and validity of the methods  and the related ethical perspectives should also be considered.

Details related to this are now in the discussion section (limitations paragraph).

Results

The results are described well and they are interesting.

Discussion

Main results are clearly presented and justified by previous studies. The meaning of the study has been discussed. There has been presented some thoughts about the reliability and validity of the study, but it could be more in-depth to look at qualitative justification criteria (credibility, transferability, dependability, confirmability, see e.g. Lincoln, Y.S. & Guba, E.G. (1985). Naturalistic Inquiry. Newbury Park, CA: Sage Publications;Lincoln, Y.S.(1995). Emerging Criteria for Quality in Qualitative and Interpretive Research. Qualitative Inquiry, 1(3),275-289. http://ideanetworking.com.au/docs/interpretiveresearchmethods/Interpretive_Criteria-1995-Lincoln-275-89.pdf)

We have added some specific elements of quality related to qualitative research. The limitations paragraph now reads “Though it is recommended that sample sizes for focus groups remain small, this study included a total of 14 participants and was a voluntary convenience sample. Many participants were proponents of FDK access, some even suggested making it mandatory, with no opinions shared that were in opposition to FDK. Yet previous research has found both proponents and opponents of FDK. It is unclear if the sentiment surrounding full-day programs is shifting, or if our findings were due, in part at least, to a sampling bias whereby those most passionate about education or FDK agreed to participate in the study. It is possible that current findings may not be generalizable to a larger population. The issues surrounding FDK access and funding span statewide, but our findings are just those from stakeholders in Southern Nevada. It is unknown if findings would have differed if focus groups were conducted state wide. Additionally, perceptions and priorities related to the education-health link may differ outside of the USA. Thus, it is possible that the transferability of current findings may not be relevant to other contexts [22]. Validity and reliability are important in qualitative research; to ensure dependability [22], this study relied on focus group methodology to collect data and an accepted method of content analysis for application, all of which are documented in detail in this manuscript. To enhance reliability and credibility [22], two researchers employed the same method of analysis across focus group content, and both coders discussed and mitigated discrepancies.  While focus groups are one methodology of collecting experiential data, it may be that individual interviews with stakeholders would have revealed experiences that were not part of the focus group discussion.”

The tables are mainly ok, but an explanation for the abbreviation NV is missing from the Table 2. It should be added.

We have added the NV abbreviation to the key below the table.

References are well chosen.

Reviewer 2 Report

Abstract and Keywords:

What about full-day kindergarten as a keyword?

Introduction:

(72) should insert number "1" before text "National Center for Education Statistics (NCES), 2014;"

Methods:

(96) "it was likely that the number was large" - approx. how many?

(98) Need to explain the meaning of "RSVPs" - Répondez, s'il vous plaît

Discussion:

(307) "from that of Elicker and Mathur (1997)" - this also needs a proper citation - (6)

Author Response

Abstract and Keywords:

What about full-day kindergarten as a keyword?

Thank you for the suggestion. We added it.

Introduction:

(72) should insert number "1" before text "National Center for Education Statistics (NCES), 2014;"

Thank you, we’ve added the 1.

Methods:

(96) "it was likely that the number was large" - approx. how many?

Unfortunately the authors are unable to speculate beyond the mention of the number of students enrolled in the Clark County School District. “Clark County School District is the 5th largest in the nation, with over 320,000 enrolled youth”

(98) Need to explain the meaning of "RSVPs" - Répondez, s'il vous plait

We reworded this to avoid the necessary explanation. It now reads “There were 18 confirmative responses to the event, and a total of 14 participants who ultimately attended the event representing stakeholders that included parents, current and former teachers, and community members.”

Discussion:

(307) "from that of Elicker and Mathur (1997)" - this also needs a proper citation - (6)

The proper reference is now added.

Reviewer 3 Report

Reviewer's report

Title: A qualitative study on parental and community stakeholder views of the link between full-day kindergarten and health in Southern Nevada

Journal: Children

This study examined what parents and other community stakeholders thought about full-day kindergarten and health in Southern Nevada. Also, the authors examined parents and other community stakeholders` priorities regarding which health issues could be improved with expanded access to full-day kindergarten.It fits well within the scope of the journal and addresses some  links between social determinants of health (early childhood care and education) and different accessibility of kindergarten.

In my opinion, the article would  benefit, if authors consider following:

 In the abstract of the article, there are two similar sentences, and one should be deleted. Participants were in favor of expanding access to FDK and felt that FDK could improve social skills, increase the amount of physical activity, and provide additional time to detect learning disabilities. Participants were overall in favor of  expanding access to FDK and felt that FDK could improve social skills, increase the amount of physical activity, and provide additional time for educators to detect additional learning disabilities when compared to half-day programs.

In general, I think this was a carefully conducted study, and quite a good example of qualitative research, which need to be named as thematic analysis. Although it reads more like qualitative description than a phenomenological study,  the explication of method is particularly well done.

However, the authors need to consider two more points. First, it is not entirely clear to me the importance of pointing out the parents were "invited to bring their children" (line 108-109).  Second, the authors may need to clarify what kind of "brief explanation of the link between education and health " was given to participants before the focus group discussion. In line 115 it is stated only that  short and long term health effects were included.

In the Result section the authors need to be more precise and use the term "parents and other community stakeholders", instead of "education stakeholders", because parents could be considered health stakeholders as well as education stakeholders. This is particularly important in the Table 2.

In one transcript of the respondent answers (lines 193-194) we are missing the information about the group.

With regard to the findings of health impact of FDK, these findings seem reasonable, although they are not surprising. However, there are only some social determinants of health that were investigated so, the authors should revise the second aim specifically pointed to the social and behavioral aspects of health.

The generalization of the findings from qualitative research is an issue as well as the transferability, so the authors need to be thoughtful about these in the conclusions. A similar study carried out in another country may raise very different responses, as they pointed out in the limitation. In addition, putting the conclusion as a separate section, would enhance the  structure quality of the article.

This paper deserves to be considered for publication, after these minor revisions. I hope the authors will find these comments useful to revise the paper and improve the quality of the article.

Author Response

Reviewer 3

Title: A qualitative study on parental and community stakeholder views of the link between full-day kindergarten and health in Southern Nevada

Journal: Children

This study examined what parents and other community stakeholders thought about full-day kindergarten and health in Southern Nevada. Also, the authors examined parents and other community stakeholders` priorities regarding which health issues could be improved with expanded access to full-day kindergarten. It fits well within the scope of the journal and addresses some  links between social determinants of health (early childhood care and education) and different accessibility of kindergarten.

In my opinion, the article would  benefit, if authors consider following:

 In the abstract of the article, there are two similar sentences, and one should be deleted. Participants were in favor of expanding access to FDK and felt that FDK could improve social skills, increase the amount of physical activity, and provide additional time to detect learning disabilities. Participants were overall in favor of  expanding access to FDK and felt that FDK could improve social skills, increase the amount of physical activity, and provide additional time for educators to detect additional learning disabilities when compared to half-day programs.

Thank you, we have deleted the duplicative sentence.

In general, I think this was a carefully conducted study, and quite a good example of qualitative research, which need to be named as thematic analysis. Although it reads more like qualitative description than a phenomenological study,  the explication of method is particularly well done.

Thank you. We have added “conventional content analysis” to the abstract and the methods (section 2.4)

However, the authors need to consider two more points. First, it is not entirely clear to me the importance of pointing out the parents were "invited to bring their children" (line 108-109). 

The authors were attempting to explain the steps that were taken to minimize barriers to attendance. We’ve added that to enhance clarity. It now reads “The focus group was held after hours (5:45pm) at a local preschool. Attempts were made to enhance convenience and minimize barriers to attendance. The preschool was centrally located in town, was accessible by public transit, and free parking was available. Parents were invited to bring their children as a replacement for the need to arrange for childcare.”

Second, the authors may need to clarify what kind of "brief explanation of the link between education and health " was given to participants before the focus group discussion. In line 115 it is stated only that  short and long term health effects were included.

The authors tried to clarify this and added the following: “A brief explanation of the link between education and health was given, which included some examples of short and long term health effects (i.e. physical education augmenting physical activity behaviors, and educational attainment resulting in fewer negative health behaviors such as smoking).”

In the Result section the authors need to be more precise and use the term "parents and other community stakeholders", instead of "education stakeholders", because parents could be considered health stakeholders as well as education stakeholders. This is particularly important in the Table 2.

This has been rephrased.

In one transcript of the respondent answers (lines 193-194) we are missing the information about the group.

This has been corrected.

With regard to the findings of health impact of FDK, these findings seem reasonable, although they are not surprising. However, there are only some social determinants of health that were investigated so, the authors should revise the second aim specifically pointed to the social and behavioral aspects of health.

The authors have tried to clarify this by adding the following sentence to the introduction (last paragraph) “Education is one of several, though not the only, social determinant of health.” 

The generalization of the findings from qualitative research is an issue as well as the transferability, so the authors need to be thoughtful about these in the conclusions. A similar study carried out in another country may raise very different responses, as they pointed out in the limitation. In addition, putting the conclusion as a separate section, would enhance the structure quality of the article.

 We have added the limitation and specifically called out transferability in the limitations section. Also, we created an additional subheading for the conclusion – thank you for the suggestion.

This paper deserves to be considered for publication, after these minor revisions. I hope the authors will find these comments useful to revise the paper and improve the quality of the article.

Yes, very much so. Thank you for your time.